# Metabolomics Point out the Effects of Carfilzomib on Aromatic Amino Acid Biosynthesis and Degradation

**DOI:** 10.3390/ijms241813966

**Published:** 2023-09-12

**Authors:** Ioanna Barla, Panagiotis Efentakis, Sofia Lamprou, Maria Gavriatopoulou, Meletios-Athanasios Dimopoulos, Evangelos Terpos, Ioanna Andreadou, Nikolaos Thomaidis, Evangelos Gikas

**Affiliations:** 1Laboratory of Analytical Chemistry, Department of Chemistry, National and Kapodistrian University of Athens, 15771 Athens, Greece; ioannamprl@chem.uoa.gr (I.B.); ntho@chem.uoa.gr (N.T.); 2Laboratory of Pharmacology, Department of Pharmacy, National and Kapodistrian University of Athens, 15771 Athens, Greece; pefentakis@yahoo.com (P.E.); sophialambrou7@gmail.com (S.L.); jandread@pharm.uoa.gr (I.A.); 3School of Medicine, Department of Clinical Therapeutics, National and Kapodistrian University of Athens, 11527 Athens, Greece; mgavria@med.uoa.gr (M.G.); mdimop@med.uoa.gr (M.-A.D.); eterpos@hotmail.com (E.T.)

**Keywords:** carfilzomib, drug-toxicity, metabolomics, post hoc analysis, HRMS, aromatic amino acid pathways

## Abstract

(1) Carfilzomib (Cfz) is an antineoplastic agent indicated for the treatment of multiple myeloma. However, its beneficial action is attenuated by the occurrence of cardiotoxicity and nephrotoxicity as the most common adverse effects. Presently, there is well-established knowledge on the pathomechanisms related to these side effects; however, the research on the metabolic alterations provoked by the drug is limited. (2) An in vivo simulation of Cfz-induced toxicity was developed in (i) Cfz-treated and (ii) control mice. An RP-HRMS-based protocol and an advanced statistical treatment were used to investigate the impact of Cfz on the non-polar metabolome. (3) The differential analysis classified the Cfz-treated and control mice and resulted in a significant number of identified biomarkers with AUC > 0.9. The drug impaired the biosynthesis and degradation of aromatic amino acids (AAA) and led to alterations of uremic toxins in the renal and urine levels. Furthermore, the renal degradation of tryptophan was affected, inducing its degradation via the kynurenine pathway. (4) The renal levels of metabolites showed impaired excretion and degradation of AAAs. Cfz was, finally, correlated with the biosynthesis of renal dopamine, explaining the biochemical causes of water and ion retention and the increase in systolic pressure.

## 1. Introduction

Carfilzomib (Cfz) is an antineoplastic agent employed for the cure of relapse/refractory multiple myeloma; however, its action is compromised by high percentages of cardiovascular and renal side effects. Previous studies have shown that Cfz nephrotoxicity is caused by the activation of the mineralocorticoid receptor (MR) through the dysregulation of water/ion transport and the urine electrolyte balance [1]. The existing metabolomics studies show that Cfz leads to metabolic dysregulations that are mainly located in the kidneys and urinary system [1,2]. Furthermore, the found metabolites prove the correlation between the drug administration with conditions of inflammation, kidney injury, and oxidative stress [2]. However, the knowledge regarding the influence of Cfz on the metabolic pathways of the circulatory and the urinary system remains limited.

Metabolomics are considered as informative tools in the drug-induced toxicity field of research. Thus, the conjugation of different metabolomics platforms, i.e., reversed-phase (RP) and HILIC chromatography, or positive and negative ionization, ensure complementarity regarding the obtained information. The current study aimed to enrich the existing knowledge by employing RP-HRMS conditions to emphasize the impacts of Cfz on the non-polar metabolome (as those compounds have a better chromatographic performance in RP-columns). The final results confirmed the already-known conclusions, but also showed that the non-poral metabolites exhibit different regulation patterns between the circulatory and urinary systems, in contrary to the polar ones. The identified metabolites showed a pronounced impact of Cfz on the renal aromatic amino acids, which are strictly linked to kidney disfunction. Also, the regulation of renal dopamine was altered, providing new evidence on the background of Cfz-induced nephrotoxicity.

## 2. Results

The library-based peak-picking workflow (using the in-house DB of metabolites internal standards, enriched with metabolites already linked to Cfz nephrotoxicity) managed to detect 246 and 140 metabolites for ESI+ and ESI−, respectively, using lists of 270 and 219 suspect metabolites accordingly. These metabolites were combined with the library-free resulted features (mz_rt) and were subjected to univariate and multivariate statistical analysis.

The PCA analysis (Figure 1a,c,e) described that, in all plasma, kidney, and urine datasets, the separation trends are in accordance with Cfz administration. However, the data from the plasma and kidneys exhibited more pronounced separation, with lower group inner variance, in contrast to the urine datasets. Additionally, PLS-DA (Figure 1b,d,f) confirmed that the developed models succeeded in classifying the two groups efficiently. The test of permutations, the misclassification test, and the ROC curve analysis of the models’ sensitivity/specificity were used to assess the validity of the prediction ability, as well as to check for overfitting. The results of these trials are summarized in Table 1.

A high number of variables were found to be differentially regulated (either up or down) due to the administration of the drug. The heatmaps of variables regulation are provided in Appendix A. It is worth mentioning that, in the urine case, several features were only detected in the Cfz samples in a reproducible way.

Afterwards, univariate ROC analysis, applied in raw data, was used to detect potential biomarkers capable of classifying a sample as Cfz-induced nephrotoxicity. The AUC values > 0.9 and VIPs > 1.5 were used as criteria for variable ranking. The urine dataset provided the higher number of differentiating variables (565), followed by the kidneys (191), and the plasma (150). A total of 65% of these important variables were found to be increased in the plasma and kidney of the Cfz-treated mice, whereas 65% of them were decreased in the urine of the same mice. 

### 2.1. Post Hoc Analysis

An increase in metabolites in the plasma and kidney of the Cfz-treated mice and a decrease in the urine of the same mice were found. This was further investigated as a potential pattern of metabolite regulation, employing ASCA and MEBA. It is important to note that, in this step, the major goal was to investigate the holistic impact of the drug in the circulatory (plasma and kidney) and urinary (kidney and urine) systems, recognizing the kidneys as a connection point. The ASCA was used to look for important interactions between the drug influence and the biosample. The major patterns resulting from the ASCA procedure regarding the differentiation of the variables in the control and Cfz mice (Figure 2a) show that there are interactions between the phenotype, i.e., Cfz/control, and the biosample, as their patterns intersect with each other at the renal-urinary level. This confirmed that the drug triggered different types of metabolic alterations in each system: corresponding impact in the blood and kidneys and reverse impact in the urine. Particularly, the levels of metabolites were equivalently increased in the plasma and kidney and decreased in the urine of the Cfz-treated mice. The leverage and SPE score (Figure 2b) were considered to detect the variables that fit the ASCA-given major patterns: 38 and 2 for the ESI+ and ESI−, respectively. In the case of the MEBA analysis, the flow of metabolites from the blood to the kidney and their concomitant excretion in urine was considered as a trajectory, connecting the circulatory and urinary systems. Thus, the plasma, kidney, and urine were considered as temporal profiles, and the Cfz and control phenotypes as the different conditions. Therefore, the MEBA time series was applied to investigate the existence of variables that were significantly influenced by the administration of Cfz in one temporal profile, e.g., the metabolite levels increased only in the kidney. The variables were evaluated using the Hotelling-T2 test, with values > 30 being statistically significant. Only two of the MEBA-proposed variables were also found to differentiate by the other statistical tests, i.e., 212.8437_1.23 was a priority variable in both the plasma (+) and urine (+), and 631.2701_8.15 in the plasma (+).

### 2.2. Identification

For the identification of the library-free protocol, the experimental data, i.e., the pseudoMSMS created by MSDIAL, employing DIA data, were compared with reference MS/MS and in silico MS/MS data, provided by online libraries, e.g., HMDB (Figure 3). Both peak-picking methods identified 152 statistically important metabolites, 57 of which resulted from the library-based approach (Table 2), using the ‘suspect-screening list’. The rest resulted from the identification procedure of the unknown features (Table 3).

## 3. Discussion

This study aimed to enrich the existing knowledge on Cfz-induced nephrotoxicity by the metabolomics prospective, performing RP chromatographic analysis, as a complementary step to Barla et al.’s published workflow [1]. Metabolomics deals with thousands of compounds, therefore, the combination of different separation methodologies offers more opportunities for ‘good performance’ for a larger number of metabolites. In other words, compounds that co-eluted in RP can be separated using HILIC. The current RP-based study confirmed the HILIC-based observations regarding the strong impact of Cfz on the circulatory and urinary systems and, furthermore, provided new evidence. The first important thing to note is that the non-polar metabolome (RP data) provided clearer classification trends, concerning PCA, between the Cfz and control mice in all datasets. This indicated that Cfz affects the non-polar metabolome in a more intense way. Also, the RP dataset led to the identification of more than 90 significantly differentiating metabolites, while the HILIC protocol identified 53. Thus, RP is considered a more informative approach.

In the current project, library-free and library-based methodologies were combined for metabolite detection. It is worth mentioning that the metabolites provided by the library-based search (red-marked loadings in Figure 4) showed a limited discriminating ability compared to that of the library-free protocol. The library-based approach showed limitations regarding the interpretation of the Cfz impact, as more than 700 statistically significant features remained unknown. Specifically, 17% of the overall important features were finally identified, with 6% being identified by the library-based protocol. The library-free methods are more informative, as they are not limited to the already-known compounds, i.e., usually amino acids, carnitines, and other naturally occurring compounds. In addition, they focus on the differentiating statistically important metabolic evidence, whether known or not, providing the chance to record features of higher specificity for the examined condition. In the current case, thirteen of the identified metabolites were detected as products of the metabolites’ metabolism, e.g., as acyl derivatives, generated by enzymatic actions on metabolites.

### 3.1. Patterns of Cfz-Induced Metabolic Regulation

It was observed that, for the Cfz-treated mice, 65% of the metabolites increased their levels in the plasma and kidneys, and 65% decreased their levels in the urine excretes. The latter suggested two opposite phenomena: (i) an increase in metabolites in the blood circulation and (ii) a decrease in them in excretes. This observation is in accordance with what we have previously shown. Cfz leads to water and salt retention via mineralocorticoid receptor (MR) in vivo activation in the kidneys, and in patients with Cfz-related renal adverse events as well. This pathomechanism was further supported, as Eplerenone administration, a clinically applicable MR blocker, reversed Cfz’s nephrotoxicity in vivo, as shown by Efentakis et al. [1]. The current data support these findings, and the increase in renal and circulatory metabolites can be attributed to the impairment of the renal reuptake mechanisms. Aiming to find patterns that are representative for the majority of metabolites, and in contrast to the previously applied univariate approach, the current study developed two multivariate approaches, i.e., ASCA and MEBA. Therefore, ASCA enables the detection of interactions between the drug and the circulatory/urinary system as well, besides estimating the significance of the main effects. Additionally, the MEBA step of the proposed workflow aimed to distinguish the variables that exhibit significant dysregulation at only one single point by assigning this “behavior” to their temporal profile. However, MEBA did not afford a significant number of *single-point dysregulated metabolites*, in contrary to the univariate approach that was applied in the HILIC data (Figure 5). A significant number of statistically important polar metabolites (from the HILIC data) were found to be accumulated only in the kidneys, while the majority of non-polar lipophilic substances (from RP data) did not show a single dysregulation peak in the temporal profiles, confirming the ASCA findings.

The kidneys perform filtration, reabsorption, secretion, and metabolism; thus, renal dysfunction affects the circulating metabolites in different, even controversial, ways [3,4]. The abnormal accumulation of specific metabolites could either result from the kidneys’ inability to catabolize substances, from extensive renal biosynthesis, or from abnormalities in reabsorption/excretion. Increased metabolite levels in the blood could be caused by their limited renal clearance as well, though, it is interesting that the polar metabolites (HILIC data) were mainly accumulated in the kidneys, while their lipophilic counterparts were increased in both the blood and the kidneys. This betrays perturbation of the metabolic transportation pathways. The existing data are in accordance with the following two facts: (i) Cfz affects renal osmolarity, leading to alterations of the ions’ exchange routes in the kidneys; and (ii) Cfz activates the MR, leading to the expression of proteins that regulate the ion/water transportation. Potentially, the latter inhibits the renal transformation of lipophilic molecules into polar derivatives, which are easily excreted by the glomerular filtration barrier. This obstructs normal renal clearance, explaining the corresponding elevated levels of non-polar metabolites in the blood.

### 3.2. Biomarkers of Cfz Nephrotoxicity

The identified metabolites were searched in the database of the European Uremic Toxin Group (https://database.uremic-toxins.org/soluteList.php, accessed on 15 May 2023), and creatine (↑ Cfz-kidneys), hypoxanthine (↑ Cfz-kidneys), N6-methyladenosine (↓ Cfz-kidney), argininic acid (↑ Cfz-urine), and 1-methyladenosine (↑ Cfz-urine) were identified as uremic biomarkers. With the exception of n6-methyladenosine, their levels were found to be increased, as in the uremic conditions. Regarding the rest of the metabolites, the following was found:Nine metabolites participate in the tryptophan (Trp) degradation pathway, i.e., tryptophan, 3-indole-propionic acid, 1H-indole-3-carboxaldehyde, 2-aminomuconic acid semialdehyde, 5-hydroxykynurenamine, indole, indoleacrylic acid, indolelactic acid, N′-formylkynurenine.Four metabolites are hydroxy fatty acids (HFAs), i.e., (S)-3,4-dihydroxybutyric acid, 3-hydroxydodecanoic acid, and 4-8-hydroxy-5,6-octadienoic acid were upregulated; whereas 3-hydroxyvalproic acid was downregulated.Three metabolites are FA derivatives, i.e., 3,5-tetradecadiencarnitine, hydroxypropionylcarnitine, and 3-hydroxyglutaric acid lactone.Two metabolites are medium- and long-chain fatty acids, i.e., 10Z-heptadecenoic acid and palmitic acid were found with increased levels; and three, i.e., heptenoic acid, 7,8-dihydropteroic acid, and 3,4-methylenesebacic acid, were found to have decreased.

### 3.3. Dysregulation of Fatty Acids Oxidation

The Cfz-treated mice showed increased levels of HFAs and derivatives, showing the Cfz induced dysregulation of FA metabolism. It is known that the defective FA metabolism, i.e., uptake, formation, and degradation, provokes severe dysregulation of renal function. The kidneys’ filtration–reabsorption operation has a high demand for energy, provided mainly by fatty acid oxidation (FAO) [5,6]. FAO is accomplished by mitochondrial β-oxidation, peroxisome β-oxidation, or microsomal ω-oxidation. Short- and medium-chain FA are preferably metabolized by mitochondria, whereas long-chain FA by peroxisomes. Peroxisomes break down the FA chain, and their shorter products are moved into the mitochondria to complete the FAO procedure [5,7]. The 3-HFAs are substrates for peroxisomal β-oxidation, and their increased levels may enhance this procedure, leading to the augmented formation of reactive oxygen species byproducts. Τhe inhibition of PPARa expression also limits the production of short acyl-CoA, which are metabolized in the mitochondria to generate ATP. The latter case indicates mitochondrial dysfunction. These two scenarios could be described by the observed increase in HFAs in the kidneys. The peroxisomal β-oxidation is regulated by the PPARa receptor, which is expressed in the kidneys, and its dysregulation has been related to acute kidney injury (AKI) [5,7]. Previous results of molecular analysis have proven that Cfz prohibits the phosphorylation of adenosine-monophosphate-activated protein kinase (AMPK) [3], making this connection the most prevailing scenario. AMPK regulates the levels of circulating free FAs by (i) activating the FAO procedure, as AMPK elevates the activity of carnitine palmitoyltransferase-1 (CPT-1); and (ii) by inhibiting lipolysis and lipogenesis [8]. Hence, in the current case, the increase in HFA levels betray limitations in CPT-1 activity.

### 3.4. Dysregulated Metabolic Pathways

The identified metabolites (Table 2 and Table 3) were subjected to pathway analysis, revealing significant alterations in the pathways of aromatic amino acid (AAAs) regulation. Specifically, the affected pathways were as follows: (i) AAA biosynthesis; (ii) the phenylalanine (Phe) metabolism; and (iii) the tryptophan (Trp) metabolism (Figure 6). It is important to note that the pathways of AAA biosynthesis and Phe metabolism were dysregulated in both the kidneys and the urine of the Cfz-treated mice, but in opposite directions, i.e., upregulated in the kidneys and downregulated in the urine (Figure 7). Trp metabolism was only altered at the renal level, showing an increase in the Cfz-treated mice.

### 3.5. Dysregulation of Biosynthesis and Metabolism of Phenylalanine

The regulation of AAAs (Phe, Tyr, and Trp) is crucial for normal kidney function [9]. The levels of the circulatory AAAs, except of Trp, seem to be negatively correlated to the glomerular filtration rate (GFR), as their levels are increased in limited GFR patients, showing limited kidney uptake [9]. However, in the current case, the opposite condition is observed. The levels of Phe, Tyr, and Trp remain unchanged in the blood of the Cfz-treated mice, whereas they are found to be increased in the kidneys and decreased in the urine. Despite their elevated renal levels, their catabolic products (phenylethylamine, 2-hydroxyphenylactate, etc.) are not altered in the kidneys of the Cfz-treated mice, indicating inhibition of AAA degradation. On the other hand, AAAs and their catabolic products decreased in the urine, confirming that the AAAs are retained by the kidneys.

In addition, AAAs are precursors of uremic toxins, e.g., indoxyl-sulfate [10]. Thus, their elevation, combined with the increased uremic toxins, e.g., hypoxanthine (↑ Cfz-kidneys), argininic acid (↑ Cfz-urine), and 1-methyladenosine (↑ Cfz-urine), strengthens the suggestion of the establishment of early uremic conditions due to the administration of the drug.

The decrease in L-Dopa and tyramine in the urine is an interesting finding, as they are precursors of dopamine. The decarboxylation of circulatory L-Dopa is the main source of dopamine in the kidneys, and the renal dopamine represents the main source of urine dopamine. It was hypothesized that the decrease in urine-excreted L-dopa reflects the corresponding decrease in renal L-Dopa and, consequently, the impairment of renal dopamine biosynthesis. As the compound is a regulator of the water and electrolyte balance, facilitating their excretion [11], the renal dopamine was further investigated. Despite the fact that the compound had not passed the statistical thresholds (AUC > 0.9, VIP > 1.5, FDR *p* value < 0.05), and, therefore, was not considered as a potential biomarker, its levels were found to be decreased in the kidneys of the Cfz-treated mice (*p* value = 0.046), but not altered in the urine (*p* value = 0.34). Overall, the dopamine was detected as decreased in both the kidneys and urine of the Cfz mice, and its precursor, L-Dopa, was detected as decreased only in the urine of the Cfz mice. This result could indicate the biochemical background of the increased systolic blood pressure that occurs as an adverse effect of Cfz [1], due to decreased vasodilation in the kidneys, as well as impairment of hormonal regulation.

### 3.6. Dysregulation of Tryptophan Metabolism

There are three pathways of Trp catabolism: the serotonin pathway, the kynurenine pathway, and the indole pyruvate pathway, which involves microbial metabolism [12]. These pathways lead to either products that maintain their indole ring, e.g., serotonin, indoxyl-sulfate, and indolelactate, or products that break the indole ring, producing kynurenine derivatives [13]. In the current study, the levels of Trp increased in the kidneys of the Cfz-treated mice, whereas the kynurenine derivative, 5-hydroxykynurenamine, increased in the plasma and kidneys, while 5-hydroxy-L-tryptophan was found significantly increased in kidneys of the Cfz-treated mice. Regarding the identified indole derivatives, no specific regulation pattern can be found between the Cfz and control groups, as the 3-indolepropionic acid, indole, and indoleacrylic acid levels are elevated, whereas the 1H-indole-3-carboxaldehyde, 3-methylene-indolenine, and indole lactic acid levels are decreased. The indole lactic acid detected in both the kidneys and urine of the Cfz-treated mice appears to be downregulated. Trp downregulation suggests an increase in its degradation rate via the kynurenine pathway, a condition linked to the developed stages of renal disease [14]. The final product of this route is NAD, however, neuroactive compounds also appear as intermediate products, responsible for the formation of free radicals [14]. The indole products of Trp are mainly formatted by microbiota, and contribute either positively or negatively, as anti-inflammation factors, or worsen the kidney-dysfunction conditions, as happens with indoxyl-sulfate, the final product of the indole pathway [12]. Most of the metabolites involved in Trp metabolism were detected in the kidneys. The significant dysregulation of Trp catabolism was statistically confirmed by pathway analysis, showing an Impact value of 0.41 (pathway topology analysis) and an FDR *p* value lower than 0.002 (enrichment analysis) (Figure 8).

#### 3.6.1. Kynurenine Sub-Pathway of Trp Catabolism

The kynurenine pathway has been shown to play an important role in the progression of kidney-related disease [15]. The renal dysregulation, as a pathway of cardiovascular disease development, is also closely related to the Trp metabolism through the kynurenine pathway [16]. A rather neglected metabolite of the pathway that seems to play a significant role in inflammatory response is 5-Hydroxykynurenamine (5-HKA). The molecule has recently emerged as the third Trp metabolism biogenic amine not involving IDO-1 [17]. In the current study, 5-HKA was increased in the plasma and kidney, and slightly decreased in the urine of the Cfz-treated mice. Its AUC values were 0.83, 0.96, and 0.86 for the plasma, kidney, and urine, respectivley, indicating a strong correlation with the development of nephrotoxicity. There is limited literature regarding compound 5-HKA, which is a product of 5-hydroxy-tryptophan (5-HTrp), formed under the influence of indoleamine 2,3-dioxygenase [17]. 5-HKA has been reported as a serotonin receptor agonist in soft tissues, inducing analogous, but not so intense, contractile response [18]. Thus, the increased levels of 5-HKA increase the renal perfusion pressure, as serotonin does, prohibiting the normal procedure of renal autoregulation [19]. On the other hand, 5-HKA has been found to ameliorate kidney inflammation through reducing the levels of inflammatory cytokines [20], whereas treatment with this substance has shown improvement of proteinuria and nitrogen balance. Under this notion, the increase in 5-HKA in the current case could indicate the existence of an inflammatory condition. Taking under consideration that renal perfusion in the microvasculature of the glomeruli is a key determinant of blood pressure, and since we have previously shown that Cfz induces increased systolic blood pressure via MR activation and subsequent water/salt retention both in in vivo and in patients, the increase in 5-HKA might be correlated with these phenomena [1].

#### 3.6.2. Indole-Pyruvate Sub-Pathway of Trp Catabolism

Indole, indoleacrylic acid (iAcr), indole propionic acid (IPA), and indole-3-lactic acid (ILA) are microbial metabolites of Trp, participating in the indole-pyruvate pathway. It is claimed that Trp’s indole derivatives are beneficial for intestinal and systematic homeostasis [21]. Kidney dysfunction dysregulates the microbial metabolism, leading to further homeostasis issues [9]. Particularly, the abovementioned indole compounds interact with the aryl hydrocarbon receptor (AhR) and pregnane X receptor (PXR), prohibiting inflammation [12]. Interestingly, in the current study, the indole compounds were found to have increased in the kidneys of the Cfz-treated mice, except that of ILA, which was found to be downregulated in the kidneys and urine of the same mice. ILA is produced by gut microbiota, specifically from the species of *Bifidobacteria* [21]. ILA exhibits anti-inflammatory action by inhibiting the production and migration of pro-inflammatory cytokines [22]. In previous studies, the compound has been found to be increased in the urine samples of patients suffering from septic acute kidney damage [23] and liver injury [24]. It is worth mentioning that, according to Madella et al., ILA, in contrast to IPA, indole, and iAcr, presents a higher degree of interaction with AhR, whereas none of them interact strongly with PRX [12]. This suggests that the decreased levels of ILA, and, therefore, its reduced anti-inflammatory activity, cannot be counterbalanced by the increased levels of the other indole derivatives that could inhibit nephrotoxicity occurrence.

## 4. Materials and Methods

The sample preparation is included in the Supplementary Material [2]. The reversed-phase (RP) chromatographic separation was performed using an Acclaim RSL C18 Column (2.1 × 100 mm, 2.2 μm, Thermo Fischer Scientific, Dreieich, Germany). The temperature was kept at 30 °C throughout all experiments, with the aid of a software-controlled oven. The mobile phases were (A) aq. Methano, (95-5, *v*/*v*) and (B) Methanol, both containing 5 mM Ammonium Formate. For the positive ionization, the mobile phases were acidified with 0.01% Formic Acid (*v*/*v*). The gradient was set as a ramp escalation of B, from 0% to 100% in 14 min. The flow was set to 0.2 mL/min, and the injection volume to 5 μL. A Dionex UltiMate 3000 RSLC (Thermo Fischer Scientific, Dreieich, Germany) UPLC chromatographic system coupled to a Maxis Impact QTOF mass spectrometer (Bruker Daltonics, Bremen, Germany), with an Apollo ESI source, was used for the analysis. The ESI temperature was set at 200 °C, and the capillary voltage was 2.5 kV for both ionization modes. Data independent acquisition (DIA) was employed, obtaining both low-collision energy (CE), 5 V, and high-CE, 24–36 V (ramp), MS data.

Library-free and library-based methodologies were used for metabolite detection. The library-based approach used a list of known metabolites obtained from the following two sources: (i) an in-house database (DB) (ESI+ and ESI−) developed using the MSMLS (Mass Spectrometry Metabolite Library of Standards, IROA Technologies LLC, Sigma Aldrich, Steinheim, Germany), and (ii) a list of metabolites already linked to Cfz nephrotoxicity known from a previous study [2]. Regarding the DB metabolites, the information on the retention time and the fragmentation was known, whereas, regarding the Cfz-nephrotoxicity metabolites, only the *m*/*z* of the precursor ion and the adduct ions were known, and, thus, the identification was based on this and on their isotopic profile as well. The software used was TASQ Client 2.1 (Bruker Daltonics, Bremen, Germany). The untargeted peak-picking workflow employed MSDIAL 4.9.2 software, which was used for the generation of pseudo-MSMS [25] files from the DIA data as well. The peak lists of the library-free and library-based protocols were concatenated, forming a unique table; the duplicates were removed; and QC-based signal correction was performed using the QCRFSC algorithm of the StatTarget [26] GUI in the R statistical language environment.

Regarding the statistical analysis, univariate tests were performed with MetaboAnalyst 5.0 online software, using the following modules: (i) “Biomarker Analysis” for univariate ROC curve analysis [27] and (ii) “Statistical Analysis (one factor)” for FDR-corrected *t*-testing. In both cases, the data were log-transformed and auto-scaled. MetaboAnalyst 5.0 was also used for pathway analysis, using the module “Pathway Analysis (targeted)”. Furthermore, SIMCA 14.1 (MKS Umetrics, Uppsala, Sweden) software was used for the development of PCA and PLS-DA classification models. The raw data were scaled by unit variance (UV). The resulting PLS-DA models were evaluated via permutation testing, a misclassification test, and the models’ ROC curve analysis.

The post hoc analysis employed the MetaboAnalyst 5.0 module “Statistical Analysis (metadata table)” to perform ANOVA-simultaneous component analysis (ASCA) [28] and multivariate empirical Bayes statistics (MEBA) [29]. To conduct this, the plasma, kidney, and urine datasets of each ESI mode were concatenated into one matrix, log-transformed, and auto-scaled. The phenotype (Cfz vs. control) and the biosample type (plasma, kidney, or urine) were specified as the metadata factors.

Finally, for the identification of the unknown features, the mz_rt were prioritized by their VIP scores (>1.5), their AUC value (>0.7), and their FDR *t*-test *p* value (<0.05). MS-FINDER 3.52 [30] and the online databases HMDB (human), urine (human), serum (human), and LipidMAPS were used to identify the compounds. The *m*/*z* range was set at 100–1000 Da; the *m*/*z* tolerance of MS1 at 5 mDa; the *m*/*z* tolerance of MS2 at 10 mDa; the fragments relative abundance cutoff was 20%; and the isotopic tolerance was 50%. For compound annotation, both the Formula Finder and Structure Finder algorithms were employed. The MyCompoundID [31,32] (http://www.mycompoundid.org/mycompoundid_IsoMS/, accessed on 15 May 2023) online library was also used to annotate the unknown variables of the kidney (+) dataset, using the MSMS search of 1 reaction.

## 5. Conclusions

This study aimed to provide further evidence on the biochemical background of Cfz-induced nephrotoxicity, emphasizing the non-polar metabolites. The metabolomics analysis identified 93 metabolites, 5 of which belonged to the class of uremic toxins. The post hoc analysis revealed interactions between the Cfz and the circulatory/urinary system, i.e., the levels of metabolites were increased in the plasma and kidneys but decreased in the urine of the Cfz group. This finding was in contrast to previous results based on polar metabolites, which showed local increases in polar metabolites in the kidneys and a non-significant alteration in the plasma level. In the current case, the non-polar metabolites were similarly increased in the plasma and kidneys, indicating their limited renal clearance. We found increased levels of HFAs in the kidneys of the Cfz-treated mice, indicating dysregulation of FAO and mitochondrial dysfunction. The novel finding was that Cfz imposed alterations in AAA biosynthesis and degradation. The Phe metabolism was impacted in the kidneys and urine, but in opposite ways (increased in the kidneys and decreased in the urine), showing limited excretion of those metabolites in the urine. Also, the decreased levels of dopamine in the kidneys of the Cfz-treated mice imply its limited renal biosynthesis, a fact that could explain the decreased vasodilation in the kidneys, leading to increased systolic blood pressure.

## Figures and Tables

**Figure 1 ijms-24-13966-f001:**
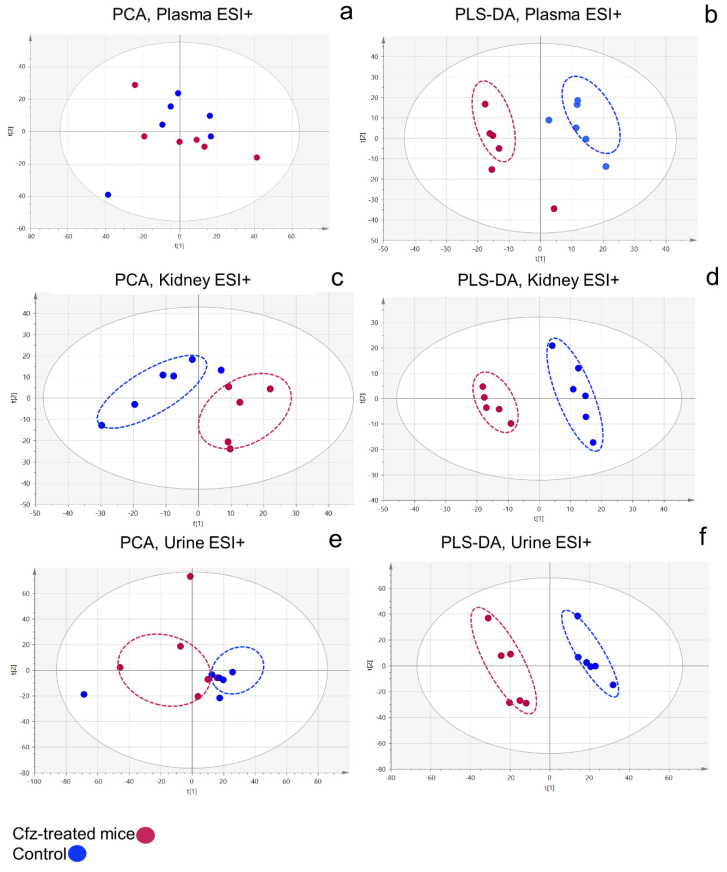
PCA scores plot of plasma ESI+ (**a**), kidney ESI+ (**c**), and urine ESI+ (**e**); and PLS-DA scores plot of plasma ESI+ (**b**), kidney ESI+ (**d**), and urine ESI+ (**f**).

**Figure 2 ijms-24-13966-f002:**
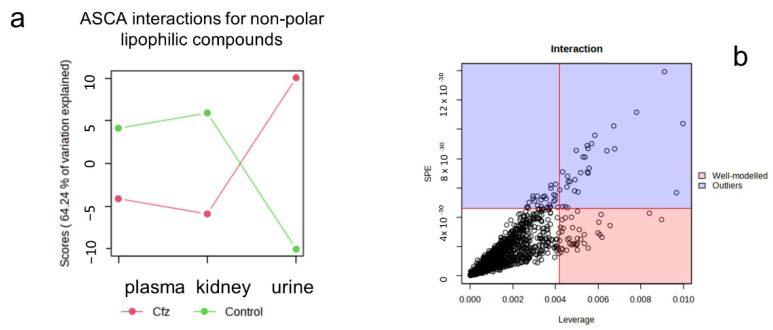
(**a**) ASCA interactions plot shows that Cfz affects the circulation in a similar way (plasma-kidney) but has opposite effect on the urinary system (kidney-urine). (**b**) Plot showing the variables exhibiting interaction due to their phenotype (Cfz/control) and the biosample type (plasma, kidney, urine).

**Figure 3 ijms-24-13966-f003:**
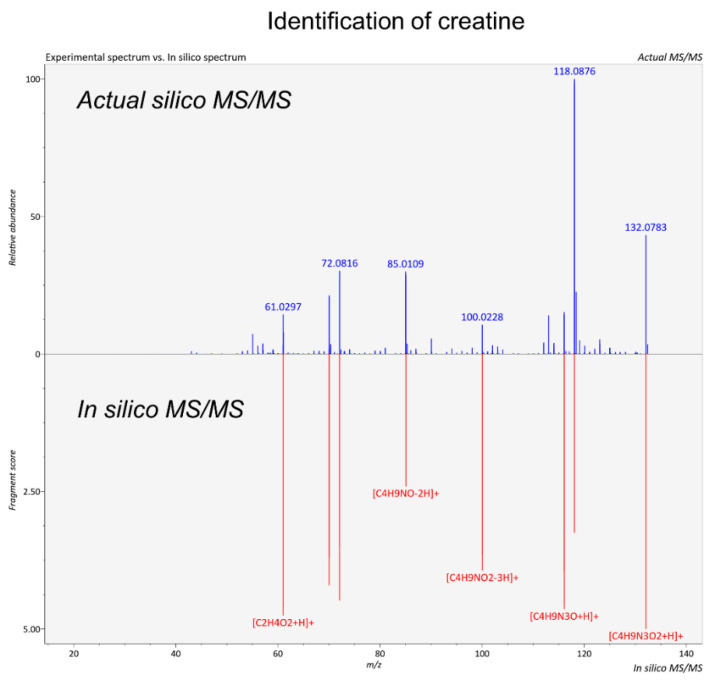
Comparison of the experimental spectrum of the unknown feature with the in silico spectrum of creatine, using MSFINDER 3.52 software.

**Figure 4 ijms-24-13966-f004:**
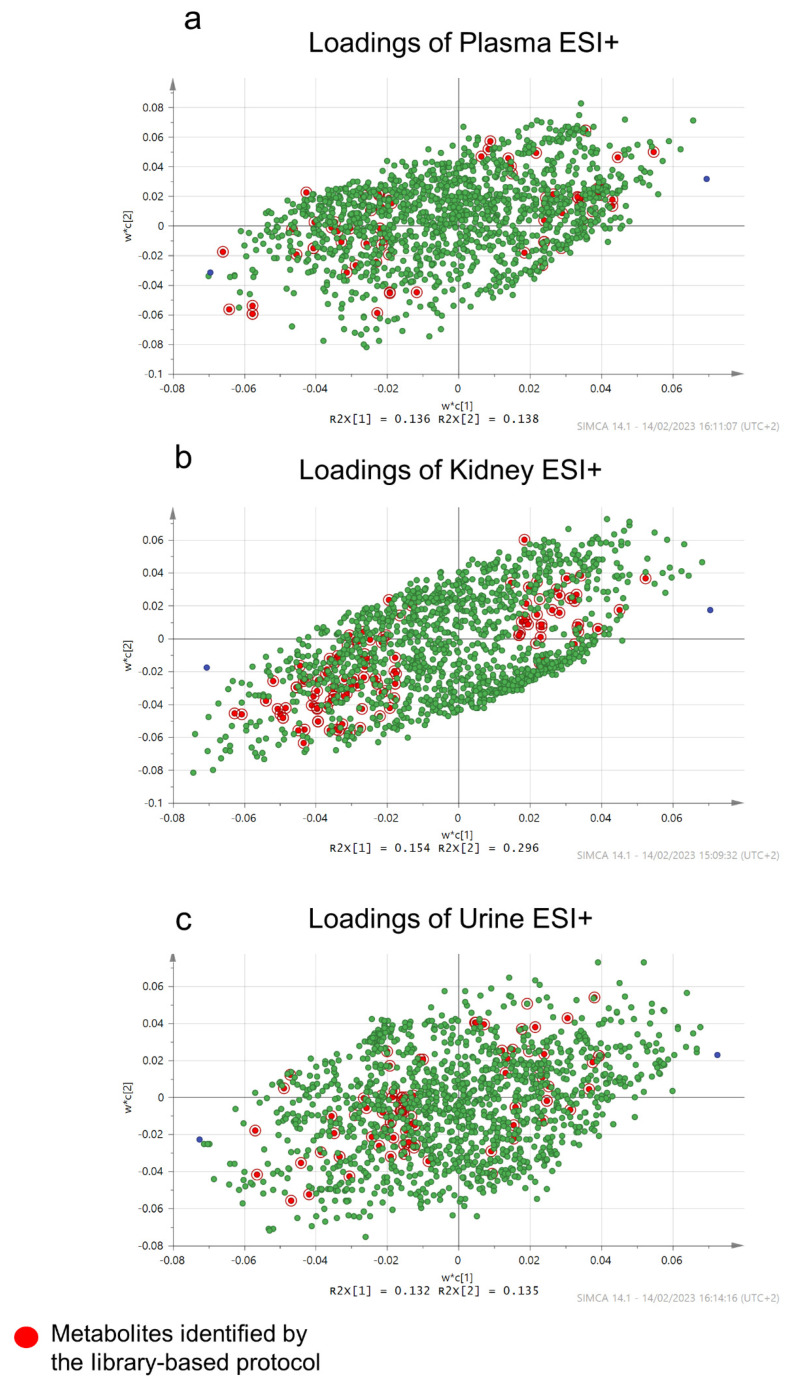
Plot of loadings of plasma ESI+ (**a**); kidney ESI+ (**b**); and urine ESI+ (**c**). The green-marked points represent the unknown variables. The red-marked points represent the metabolites detected by the library-based approach. The blue points represent the average of the y’y axes for the two groups.

**Figure 5 ijms-24-13966-f005:**
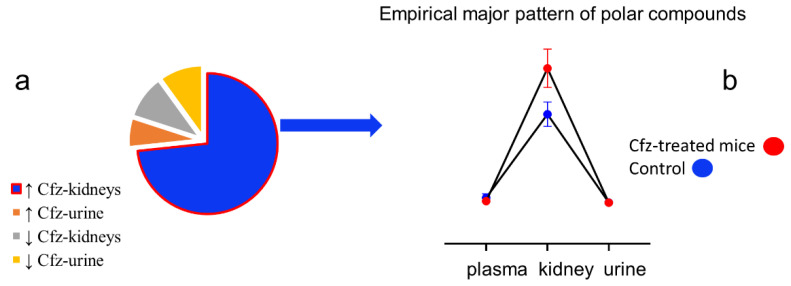
(**a**) Description of the regulation of polar metabolites (HILIC data) in the Cfz-treated mice and (**b**) empirical major pattern of polar metabolites (HILIC data).

**Figure 6 ijms-24-13966-f006:**
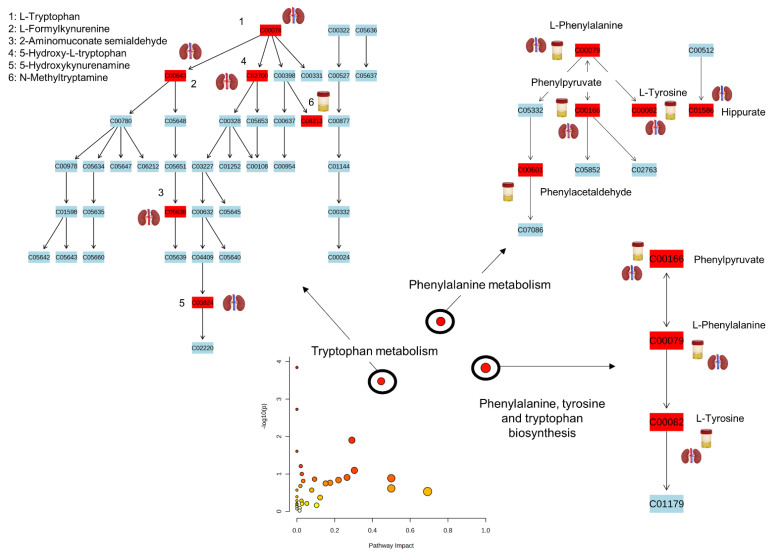
Summary of the altered metabolic pathways resulting from the pathway analysis using MetaboAnalys 5.0. In the diagram of Pathway impact-log10(p) the colors represent the importance of the pathway. The light yellow represents the pathway that was less disrupted, and the red represent the pathway that was disrupted the most.

**Figure 7 ijms-24-13966-f007:**
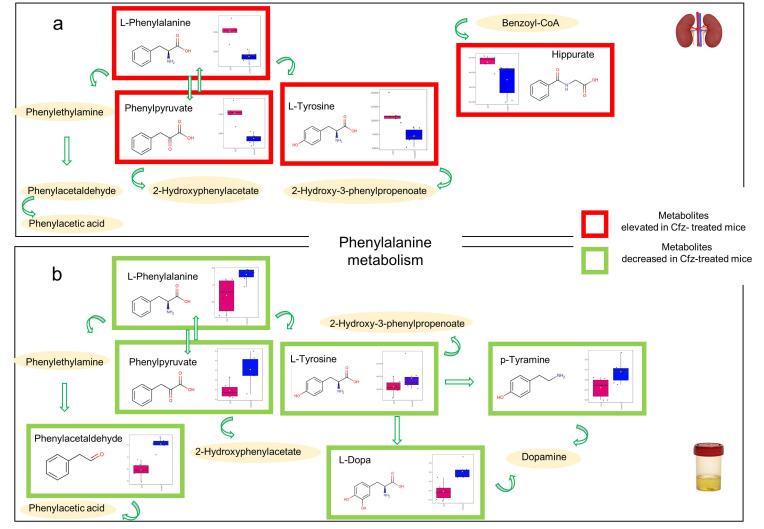
Summary of Phe metabolism in the kidney (**a**) and urine (**b**). The metabolites included in the red border were upregulated and those included in the green border were downregulated in Cfz-treated mice. The metabolites described only by their name were not affected by the drug. The box-plots represent the regulation of the metabolites in the samples: the pink represent the Cfz-treated mice and, the blue the Control mice.

**Figure 8 ijms-24-13966-f008:**
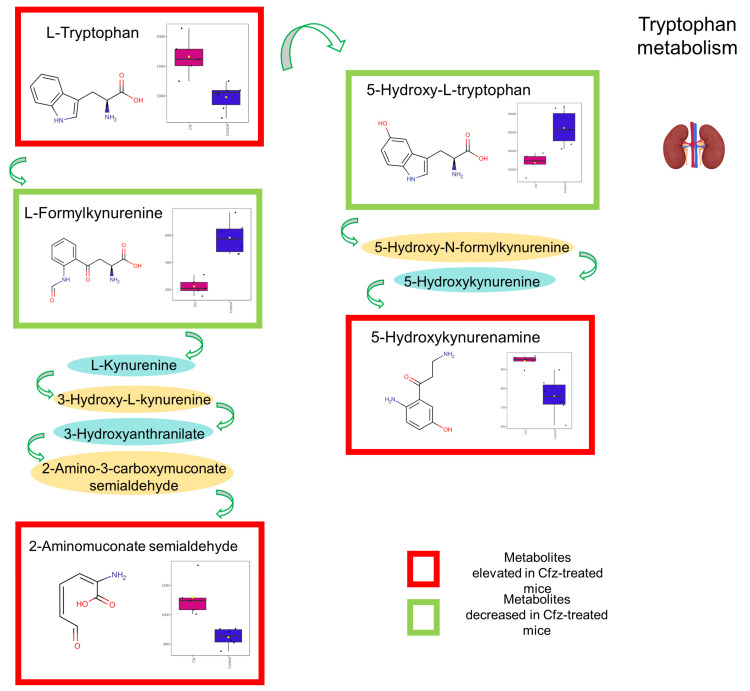
Summary of the detected alterations occurring in the Trp degradation pathway at renal level. The metabolites included in the red border were upregulated, and those included in the green border were downregulated in Cfz-treated mice. The metabolites described only by their name were not affected by the drug. The box-plots represent the regulation of the metabolites in the samples: the pink represent the Cfz-treated mice and, the blue the Control mice.

**Table 1 ijms-24-13966-t001:** Summary of results from the PLS-DA models’ performance, namely, measure of fit (R2Y), prediction ability (Q2), and the results of the permutations test (Q2permut. and R2permut.) and the misclassification error.

	R2(Y) PLSDA	Q2 PLSDA	R2(Y) PERMUT.	Q2 PEMUT.	AUC ROC’ PLSDA	Miss CLAS. Error (CFZ) (%)/PLS-DA
Plasma (+)	1	0.93	1	0.89	1	0
Kidney (+)	0.99	0.88	0.98	−0.23	1	0
Urine (+)	0.99	0.94	0.93	0.063	1	0
Plasma (−)	0.98	0.62	0.97	0.32	1	0
Kidney (−)	0.99	0.77	0.93	0.01	1	0
Urine (−)	0.99	0.82	0.93	0.13	1	0

**Table 2 ijms-24-13966-t002:** Summary of the statistically important metabolites detected by the library-based approach. The (*) reference in the “Comment” column describes those metabolites that were added in the library-based list of peak-picking as Cfz-nephrotoxicity markers, emerged in previous studies. This (↓) represents the metabolites that were downregulated in the Cfz-treated mice and the (↑) represent those metabolites that were upregulated in the Cfz-treated mice.

Dataset	Metabolite	Regulation (Cfz)	AUC	*p* Value	VIP	Comment
kidney (−)	D-(+)-Galactose	↓	0.9	0.0306	2	
kidney (−)	D-(+)-Galacturonic acid	↓	0.9	0.0186	2	
kidney (−)	D-Glucuronic acid	↓	0.9	0.0785	2	
kidney (−)	L-Phenylalanine	↑	0.8	0.8056	2	
kidney (−)	Maleic acid	↑	0.8	0.0388	2	
kidney (+)	[(S)-3-Hydroxy-N-methylcoclaurine + NH]	↑	1	0.0044	2	*
kidney (+)	[166.4847_6.96]	↑	0.9	0.0118	2	*
kidney (+)	[166.8851_6.95]	↑	1	0.0037	2	*
kidney (+)	[4-Ethylbenzoic acid + NH]	↑	0.8	0.027	2	*
kidney (+)	[86.3838_6.94]	↑	0.9	0.028	2	*
kidney (+)	[L-Tyrosine-O]	↑	0.8	0.027	2	*
kidney (+)	1-Phenylethylamine	↑	0.9	0.0065	2	
kidney (+)	2-Methylglutaric acid	↓	0.9	0.0191	2	
kidney (+)	3-Methylglutaric acid	↓	0.9	0.0191	2	
kidney (+)	4-Quinolinecarboxylic acid	↓	0.9	0.0144	2	
kidney (+)	Adipic acid	↓	0.9	0.0191	2	
kidney (+)	Cortexolone	↑	0.9	0.0143	2	
kidney (+)	DL-Normetanephrine	↑	0.8	0.027	2	
kidney (+)	Hippuric acid	↓	1	0.0062	2	
kidney (+)	L-Asparagine	↑	0.9	0.0142	2	
kidney (+)	L-Phenylalanine	↑	0.8	0.027	2	
kidney (+)	NG,NG-Dimethylarginine	↑	0.9	0.003	2	
kidney (+)	NG,NG′-Dimethyl-L-arginine	↑	0.9	0.003	2	
kidney (+)	Perillyl alcohol	↓	0.9	0.0196	1	
kidney (+)	Pyridoxamine	↑	0.9	0.0	1.8	
kidney (+)	Quinaldic acid	↓	0.9	0.0	1.6	
plamsa (+)	L-Homoserine	↓	0.8	0.2906	2	
plamsa (+)	L-Methionine	↓	0.8	0.1365	2	
plamsa (+)	L-Threonine	↓	0.8	0.2604	2	
plamsa (+)	N1-Methyl-2-pyridone-5-carboxamide	↑	0.9	0.03	2	
plasma (−)	N-Acetyl-L-leucine	↓	0.9	0.0027	2	
urine (−)	D-(+)-Galactose	↑	0.9	0.0156	1	
urine (−)	Methylmalonic acid	↓	1	0.003	2	
urine (−)	Nonanoate	↓	1	0.0019	2	
urine (+)	(R)-Salsolinol	↓	1.0	0.0	2.0	*
urine (+)	[(R)-Salsolinol + C_3_H_5_NOS]	↓	1.0	0.0	2.0	*
urine (+)	[4-Ethylbenzoic acid + NH]	↓	0.9	0.0274	2	*
urine (+)	[L-Tyrosine-O]	↓	0.9	0.0274	2	*
urine (+)	[Pipecolic acid + C_4_H_3_N_3_]	↓	0.9	0.0077	2	*
urine (+)	[p-Synephrine + NH]	↑	0.9	0.0032	2	*
urine (+)	1-Methyladenosine	↑	0.9	0.0221	2	
urine (+)	3-Amino-4-hydroxybenzoic acid	↑	0.8	0.0254	2	
urine (+)	4-COUMARATE	↓	0.8	0.0285	2	
urine (+)	4-Ethylbenzoic acid	↓	0.9	0.0108	2	
urine (+)	5-Hydroxy-L-tryptophan	↓	0.9	0.0215	2	
urine (+)	Argininic acid	↑	1	0.0245	2	
urine (+)	DL-Normetanephrine	↓	0.9	0.0274	2	
urine (+)	L-Citrulline	↑	1	0.026	2	
urine (+)	L-Phenylalanine	↓	0.9	0.0274	2	
urine (+)	L-Tyrosine	↓	0.9	0.0074	2	
urine (+)	Lumazine	↓	0.9	0.0125	2	
urine (+)	L-Valine	↓	0.9	0.0174	2	
urine (+)	N(6)-Methyllysine	↑	0.9	0.005	2	
urine (+)	N-Acetyl-5-hydroxytryptamine	↓	1	0.0012	2	
urine (+)	N-Methyltryptamine	↓	0.9	0.0192	2	
urine (+)	Salsolinol	↓	1.0	0.0	2.0	
urine (+)	Tyramine	↓	0.9	0.0	1.7	

**Table 3 ijms-24-13966-t003:** Identification table for the statistically important features resulting from the library-free peak-picking methods. This (↓) represents the metabolites that were downregulated in the Cfz-treated mice and the (↑) represent those metabolites that were upregulated in the Cfz-treated mice.

Biosample	*m*/*z*	RT	Formula	Error (mDa)	Score (MS1)	Regulation (Cfz)	Compound	Score (MS2)	Adduct Type	Library	Reaction
Kidney	205.0991	3.61	C_11_H_11_NO_2_	−0.0025	1	↑	3-Indolepropionic acid	0.93	[M + H]+	One-Reaction/MCID	[+NH]
Kidney	175.1204	1.29	C_7_H_16_NO_2_	−0.00065	1	↑	Acetylcholine	0.77	[M + H]+	One-Reaction/MCID	[+CO]
Urine	156.9912	1.72	C_4_H_8_O_4_	−1.4	4.1	↑	(S)-3,4-Dihydroxybutyric acid	7.2	[M − H]−	HMDB (MSFINDER)	-
Urine	308.1059	4.99	C_12_H_21_NO_4_S_2_	−7	3.7	↑	(S)-Succinyldihydrolipoamide	6.1	[M + H]+	HMDB (MSFINDER)	-
Urine	267.236	13.65	C_17_H_32_O_2_	−3	5.1	↑	10Z-Heptadecenoic acid	7	[M − H]−	HMDB (MSFINDER)	-
Kidney	144.0475	5.51	C_9_H_7_NO	−2.3	4	↓	1H-Indole-3-carboxaldehyde	7.4	[M − H]−	HMDB (MSFINDER)	-
Kidney	601.3402	3.31	C_27_H_53_O_12_P	−5.1	3.8	↑	1-Stearoylglycerophosphoinositol	6.3	[M + H]+	HMDB (MSFINDER)	-
Kidney	142.0492	1.31	C_6_H_7_NO_3_	0.7	3.6	↑	2-Aminomuconic acid semialdehyde	6	[M + H]+	HMDB (MSFINDER)	-
Kidney	129.0202	1.2	C_5_H_6_O_4_	−1.5	3.7	↓	2-Hydroxyglutaric acid lactone	6.3	[M − H]−	HMDB (MSFINDER)	-
Kidney	170.0618	3.64	C_8_H_11_NO	−3.97609	1	↑	2-Hydroxyphenethylamine	0.77	[M + H]+	One-Reaction/MCID	[+CO]
Urine	368.2815	10.81	C_21_H_37_NO_4_	−2.2	4	↑	3,5-Tetradecadiencarnitine	7.05	[M + H]+	HMDB (MSFINDER)	-
Urine	227.1292	4.64	C_12_H_18_O_4_	−1.5	3.8	↓	3,4-Methylenesebacic acid	6.4	[M + H]+	HMDB (MSFINDER)	-
Kidney	245.183	1.26	C_12_H_24_O_3_	−0.00881	1	↑	3-Hydroxydodecanoic acid	0.65	[M + H]+	One-Reaction/MCID	[+CO]
Urine	219.1172	3.94	C_10_H_18_O_5_	5	3.7	↓	3-Hydroxysebacic acid	6.2	[M + H]+	HMDB (MSFINDER)	-
Urine	161.1088	4.89	C_8_H_16_O_3_	−1.4	3.7	↓	3-Hydroxyvalproic acid	6.69	[M + H]+	HMDB (MSFINDER)	-
Urine	130.0665	5.58	C_9_H_7_N	−1.3	4.3	↓	3-Methylene-indolenine	6.5	[M + H]+	HMDB (MSFINDER)	-
Urine	259.0945	2.81	C_10_H_14_N_2_O_6_	−4	4.1	↓	3-Methyluridine	6.6	[M + H]+	HMDB (MSFINDER)	-
Plasma	129.0926	4.46	C_7_H_12_O_2_	−1.5	4	↓	4-Heptenoic acid	6.9	[M + H]+	HMDB (MSFINDER)	-
Kidney	409.1928	3.61	C_11_H_21_N_3_O_5_	0.000894	1	↑	5-Acetamidovalerate	0.7	[M + H]+	One-Reaction/MCID	[+C_5_H_3_N_5_]
Urine	296.067	2.09	C_8_H_14_N_3_O_7_P	−3	4	↓	5-Aminoimidazole ribonucleotide	6.9	[M + H]+	HMDB (MSFINDER)	-
Kidney	296.0681	1.24	C_8_H_14_N_3_O_7_P	−3.7	3.4	↑	5-Aminoimidazole ribonucleotide	5.4	[M + H]+	HMDB (MSFINDER)	-
Urine	181.0987	2.51	C_9_H_12_N_2_O_2_	−1.5	4	↓	5-Hydroxykynurenamine	6.5	[M + H]+	HMDB (MSFINDER)	-
Kidney	181.0879	2.26	C_9_H_12_N_2_O_2_	10	4.2	↑	5-Hydroxykynurenamine	7.2	[M + H]+	HMDB (MSFINDER)	-
Plasma	181.0874	2.27	C_9_H_12_N_2_O_2_	9.75	3.2	↑	5-Hydroxykynurenamine	5.2	[M + H]+	HMDB (MSFINDER)	-
Urine	321.045	5.61	C_10_H_15_N_2_O_8_P	4.3	4	↑	5-Thymidylic acid	6.7	[M − H]−	HMDB (MSFINDER)	-
Kidney	212.0781	1.52	C_7_H_9_N_5_O_3_	−1.4	3.7	↓	6-Carboxy-5,6,7,8-tetrahydropterin	6.2	[M + H]+	HMDB (MSFINDER)	-
Kidney	315.1123	5.19	C_14_H_14_N_6_O_3_	7.4	3.7	↓	7,8-Dihydropteroic acid	6.3	[M + H]+	HMDB (MSFINDER)	-
Plasma	157.0848	3.62	C_8_H_12_O_3_	1.1	4.3	↑	8-Hydroxy-5,6-octadienoic acid	67	[M + H]+	HMDB (MSFINDER)	-
Urine	220.065	5.12	C_11_H_9_NO_4_	−4	3.5	↓	8-Methoxykynurenate	5.5	[M + H]+	HMDB (MSFINDER)	-
Kidney	117.0587	3.62	C_5_H_8_O_3_	−4	3.7	↑	Alpha-ketoisovaleric acid	7	[M + H]+	HMDB (MSFINDER)	-
Urine	221.0701	5.49	C_7_H_12_N_2_O_6_	6	3.5	↓	Aspartyl-Serine	6.02	[M + H]+	HMDB (MSFINDER)	-
Urine	130.062	1.55	C_4_H_9_N_3_O_2_	0.4	4.7	↑	Beta-Guanidinopropionic acid	7.7	[M − H]−	HMDB (MSFINDER)	-
Kidney	149.0612	3.06	C_9_H_8_O_2_	−1.6	3.7	↑	Cinnamic acid	6.5	[M + H]+	HMDB (MSFINDER)	-
Kidney	132.083	3.62	C_4_H_9_N_3_O_2_	−5.3	3.9	↑	Creatine	8.2	[M + H]+	HMDB (MSFINDER)	-
Urine	112.0516	2.4	C_4_H_5_N_3_O	−0.8	3.5	↓	Cytosine	6.2	[M + H]+	HMDB (MSFINDER)	-
Plasma	227.0721	9.32	C_9_H_12_N_2_O_5_	−4.6	0.1	↑	Deoxyuridine	7.9	[M − H]−	HMDB (MSFINDER)	-
Urine	199.0076	4.81	C_4_H_9_O_7_P	−6.3	3.9	↑	D-Erythrose 4-phosphate	6.7	[M − H]−	HMDB (MSFINDER)	-
Kidney	240.1038	2.63	C_9_H_13_N_5_O_3_	5	4.1	↑	Dihydrobiopterin	7.6	[M + H]+	HMDB (MSFINDER)	-
Urine	433.1156	4.13	C_21_H_20_O_10_	−2.9	3.9	↑	Dihydrodaidzein 7-O-glucuronide	6.9	[M + H]+	HMDB (MSFINDER)	-
Kidney	160.1348	1.61	C_8_H_17_NO_2_	−1.6	4	↑	DL-2-Aminooctanoic acid	6.9	[M + H]+	HMDB (MSFINDER)	-
Plasma	160.1339	1.36	C_8_H_17_NO_2_	0.79	4	↑	DL-2-Aminooctanoic acid	7.23	[M + H]+	HMDB (MSFINDER)	-
Urine	229.0189	4.78	C_5_H_11_O_8_P	−6.2	4.6	↑	D-Ribose 5-phosphate	8.3	[M − H]−	HMDB (MSFINDER)	-
Urine	419.1381	4.96	C_21_H_22_O_9_	−4.4	3.6	↑	Equol 7-O-glucuronide	6.41	[M + H]+	HMDB (MSFINDER)	-
Kidney	276.1033	1.77	C_12_H_13_N_5_O_3_	5.5	3.5	↓	Ethenodeoxyadenosine	6.2	[M + H]+	HMDB (MSFINDER)	-
Kidney	457.1166	4.26	C_17_H_21_N_4_O_9_P	−3.9	4.1	↑	Flavin Mononucleotide	7.1	[M + H]+	HMDB (MSFINDER)	-
Kidney	261.148	3.72	C_11_H_20_N_2_O_5_	−3.4	4	↑	gamma-Glutamylisoleucine	6.8	[M + H]+	HMDB (MSFINDER)	-
Urine	223.1097	3.24	C_11_H_14_N_2_O_3_	−2	3.9	↓	Glycyl-Phenylalanine	6.26	[M + H]+	HMDB (MSFINDER)	-
Kidney	146.062	3.62	C_3_H_7_N_3_O_2_	−0.00653	1	↑	Guanidoacetic acid	0.23	[M + H]+	One-Reaction/MCID	[+CO]
Kidney	118.0666	3.62	C_3_H_7_N_3_O_2_	−5	3.8	↑	Guanidoacetic acid	6.5	[M + H]+	HMDB (MSFINDER)	-
Urine	140.072	4.36	C_6_H_9_N_3_O	−1.3	3.5	↓	Histidinal	5.8	[M + H]+	HMDB (MSFINDER)	-
Urine	213.0928	7.97	C_8_H_12_N_4_O_3_	5	4.1	↓	Histidinyl-Glycine	6.6	[M + H]+	HMDB (MSFINDER)	-
Urine	183.0672	1.81	C_9_H_10_O_4_	−1.8	3.9	↓	Homovanillic acid	7.3	[M + H]+	HMDB (MSFINDER)	-
Kidney	351.1743	4.96	C_10_H_19_NO_5_	3.002896	1	↓	Hydroxypropionylcarnitine	0.97	[M + H]+	One-Reaction/MCID	[+(C_5_H_5_N_5_-H_2_O)]
Kidney	137.4083	2.41	C_5_H_4_N_4_O	−1	4.2	↑	Hypoxanthine	8.4	[M + H]+	HMDB (MSFINDER)	-
Urine	157.0668	6.78	C_6_H_8_N_2_O_3_	−6	3.4	↓	Imidazolelactic acid	6.2	[M + H]+	HMDB (MSFINDER)	-
Kidney	116.0519	3.54	C_8_H_7_N	−2.9	4.1	↑	Indole	8.07	[M − H]−	HMDB (MSFINDER)	-
Kidney	331.1674	3.06	C_11_H_9_NO_2_	−0.00272	0.93	↑	Indoleacrylic acid	0.7	[M + H]+	One-Reaction/MCID	[+C_7_H_13_NO_2_]
Urine	206.0818	5.16	C_11_H_11_NO_3_	−0.5	4	↓	Indolelactic acid	6.5	[M + H]+	HMDB (MSFINDER)	-
Kidney	222.072	4.44	C_11_H_11_NO_3_	0.003536	1	↓	Indolelactic acid	0.8	[M + H]+	One-Reaction/MCID	[+O]
Urine	245.0966	6.21	C_13_H_12_N_2_O_3_	−4.8	3.8	↓	Indolylacryloylglycine	5.8	[M + H]+	HMDB (MSFINDER)	-
Urine	310.0945	3.19	C_14_H_15_NO_7_	−2.5	4.2	↑	Inodxyl glucuronide	7	[M + H]+	HMDB (MSFINDER)	-
Plasma	229.0608	2.76	C_11_H_20_N_2_O_3_	−0.9	3.6	↓	Isoleucylproline	6.8	[M+Na]+	HMDB (MSFINDER)	-
Kidney	173.1061	1.69	C_6_H_14_N_4_O_2_	−1.6	5.2	↑	L-Arginine	8.8	[M − H]−	HMDB (MSFINDER)	-
Kidney	869.4803	5.67	C_40_H_72_N_2_O_18_	−0.5	3.5	↑	Lc3Cer	6	[M + H]+	HMDB (MSFINDER)	-
Kidney	162.114	1.59	C_7_H_15_NO_3_	−1.4	4.1	↓	L-Carnitine	7.4	[M + H]+	HMDB (MSFINDER)	-
Urine	198.0775	4.36	C_9_H_11_NO_4_	−1.4	4.1	↓	L-Dopa	4.34	[M + H]+	HMDB (MSFINDER)	-
Kidney	260.9009	1.18	C_11_H_21_N_3_O_4_	8	3.8	↓	Leucyl-Glutamine	6.3	[M + H]+	HMDB (MSFINDER)	-
Urine	229.1557	1.96	C_11_H_20_N_2_O_3_	−0.63	4	↑	Leucylproline	7.2	[M + H]+	HMDB (MSFINDER)	-
Urine	322.1084	7.24	C_11_H_19_N_3_O_6_S	−0.46	4	↓	L-L-Homoglutathione	6.34	[M + H]+	HMDB (MSFINDER)	-
Kidney	203.0821	3.54	C_11_H_12_N_2_O_2_	−0.7	4.3	↑	L-Tryptophan	8.2	[M − H]−	HMDB (MSFINDER)	-
Urine	247.1092	4.95	C_10_H_18_N_2_O_3_S	1.3	3.7	↓	Methionyl-Proline	6.1	[M + H]+	HMDB (MSFINDER)	-
Kidney	264.1019	3.15	C_11_H_15_N_5_O_4_	0.007764	1	↓	N6-Methyladenosine	0.73	[M + H]+	One-Reaction/MCID	[-H_2_O]
Kidney	365.154	4.52	C_14_H_24_N_2_O_9_	2.5	4.4	↓	N-Acetylmuramoyl-Ala	7.4	[M + H]+	HMDB (MSFINDER)	-
Kidney	265.0966	3.15	C_5_H_9_NO_4_	0.007765	1	↓	N-Acetylserine	0.67	[M + H]+	One-Reaction/MCID	[+(C_5_H_5_N_5_-H_2_O)]
Urine	175.1494	7.06	C_8_H_18_N_2_O_2_	−5	3.9	↓	Ne,Ne dimethyllysine	6.41	[M + H]+	HMDB (MSFINDER)	-
Kidney	279.0938	4.01	C_11_H_12_N_2_O_4_	0.003199	0.9	↓	N′-Formylkynurenine	0.6	[M + H]+	One-Reaction/MCID	[+C_2_H_2_O]
Kidney	131.0508	3.06	C_4_H_6_N_2_O_2_	−0.00623	1	↑	N-Methylhydantoin	0.6	[M + H]+	One-Reaction/MCID	[+O]
Kidney	613.1562	3.08	C_20_H_32_N_6_O_12_S_2_	3	3.8	↑	Oxidized glutathione	6.9	[M + H]+	HMDB (MSFINDER)	-
Urine	255.5898	13.86	C_16_H_32_O_2_	0.15	4.3	↑	Palmitic acid	8.5	[M − H]−	HMDB (MSFINDER)	-
Urine	181.0662	6.79	C_7_H_8_N_4_O_2_	5	3.8	↓	Paraxanthine	6.72	[M + H]+	HMDB (MSFINDER)	-
Urine	121.0658	7.97	C_8_H_8_O	−0.8	4	↓	Phenylacetaldehyde	6.7	[M + H]+	HMDB (MSFINDER)	-
Urine	165.0564	1.85	C_9_H_8_O_3_	−1.4	3.7	↓	Phenylpyruvic acid	6.5	[M + H]+	HMDB (MSFINDER)	-
Kidney	165.0561	1.88	C_9_H_8_O_3_	−1.9	3.7	↑	Phenylpyruvic acid	6.4	[M + H]+	HMDB (MSFINDER)	-
Urine	304.1777	1.74	C_14_H_25_NO_6_	−2.2	3.9	↓	Pimelylcarnitine	6.74	[M + H]+	HMDB (MSFINDER)	-
Urine	230.1195	7.66	C_9_H_15_N_3_O_4_	−1.5	3.7	↓	Prolyl-Asparagine	5.85	[M + H]+	HMDB (MSFINDER)	-
Urine	131.1043	5.15	C_7_H_14_O_2_	2.8	3.4	↓	Propyl butyrate	5.8	[M + H]+	HMDB (MSFINDER)	-
Urine	169.0985	2.99	C_8_H_12_N_2_O_2_	−1.2	3.8	↓	Pyridoxamine	6.9	[M + H]+	HMDB (MSFINDER)	-
Kidney	201.1159	3.53	C_10_H_18_O_4_	−2.7	5.1	↓	R-2-Hydroxy-3-methylbutanoic acid 3-Methylbutanoyl	8.5	[M − H]−	HMDB (MSFINDER)	-
Urine	455.0823	3.44	C_13_H_19_N_4_O_12_P	−0.8	3.8	↓	SAICAR	6.7	[M + H]+	HMDB (MSFINDER)	-
Plasma	380.258	13.19	C_18_H_38_NO_5_P	−3	4	↑	Sphingosine 1-phosphate	6.8	[M + H]+	HMDB (MSFINDER)	-
Plasma	190.0878	6.55	C_8_H_15_NO_2_S	1.4	4	↓	S-Prenyl-L-cysteine	6.2	[M + H]+	HMDB (MSFINDER)	-
Kidney	99.0089	1.56	C_4_H_4_O_3_	−0.14	3.9	↓	Succinic anhydride	6.29	[M − H]−	HMDB (MSFINDER)	-
Urine	391.176	7.67	C_22_H_22_N_4_O_3_	0.4	3.9	↓	Tryptophyl-Tryptophan	6	[M + H]+	HMDB (MSFINDER)	-
Urine	113.0353	2.61	C_4_H_4_N_2_O_2_	−0.8	3.6	↓	Uracil	6.3	[M + H]+	HMDB (MSFINDER)	-
Kidney	113.0361	1.93	C_4_H_4_N_2_O_2_	−1	4.2	↑	Uracil	7.12	[M + H]+	HMDB (MSFINDER)	-
Kidney	147.1143	1.3	C_5_H_10_N_2_O_4_	−1.5	3.8	↑	Ureidoisobutyric acid	7	[M + H]+	HMDB (MSFINDER)	-
Urine	139.0407	7.18	C_6_H_6_N_2_O_2_	−2	3.4	↓	Urocanic acid	6.5	[M + H]+	HMDB (MSFINDER)	-

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
