# Peer review of "Metabolomics Point out the Effects of Carfilzomib on Aromatic Amino Acid Biosynthesis and Degradation"

_ijms, 2023, doi:10.3390/ijms241813966_

Round 1
Reviewer 1 Report
In this study authors describe a “Metabolomics study pointing out the effects of Carfilzomib on aromatic 2 amino acids biosynthesis and degradation.” The article is interesting from a biomedical point of view; therefore, it should interest the reader. I recommend the article for publication with minor revision. There are the following points, that should be addressed-
1 The authors need to describe, how they extracted metabolites from plasma, kidney, and urine. It is challenging to measure endogenous compounds in biological matrices due to matrix effect and ion suppression. If the author encounters any matrix effect then how they overcome it.
2 Author can provide an upregulated and down-regulated metabolites list in the supplementary data file.
3 Did the author try a heatmap to show changes in metabolites up and down regulations?
4 The authors need to describe in the discussion section of LC-MS/MS analysis, how they overcome the problem of co-eluting peaks of similar isomeric analytes in complex biological matrices.
English is fine with minor grammatical refinememnt
Reviewer 2 Report
The manuscript by Ioanna Barla et al. studied the alterations of metabolites in the plasma, kidney and urine in mice treated with Carfilzomib (Cfz), a non-reversible proteasome inhibitor approved for the treatment of patients with multiple myeloma. Cfz usage is known to be associated with cardiotoxicity and nephrotoxicity, however, its toxic effects on metabolic pathways are not clear. This study found the impact of Cfz on several metabolic pathways, particularly the biosynthesis and degradation of aromatic amino acids - phenylalanine, tryptophan and tyrosine, and fatty acid oxidation, and revealed the interactions between Cfz and non-polar metabolite levels in circulatory, renal and urinary systems. The authors also found the decreased level of L-dopa in urine of Cfz-treated mice and suggested that the impairment of dopamine biosynthesis in kidneys could explain the biochemical cause of Cfz on water and ion retention, and the subsequent increase of systolic pressure and cardiac preload. Overall, the manuscript is well written and organized, and presents the important data that give new insight into the metabolic disturbances imposed by Cfz. The manuscript is very informative in the field of Cfz-related renal and cardiovascular adverse events. Methodology used (including reversed phase chromatography, library-free and library-based methodologies used in metabolites detection, two multivariate approaches- ASCA and MEBA used in post-hoc analyses) are also appropriate. Thus, I would like to recommend the manuscript for publication.
Specific comment –
It is unclear whether the authors also studied the polar metabolites obtained from HILIC chromatography (Figure. 5)
It is not clear whether levels of L-dopa and dopamine were decreased in the kidney and urine of Cfz-treated mice.
The authors should thoroughly check the References cited in the manuscript. For example, Ref.1 and 22 are incomplete.
